# Study of Residual Stresses and Austenite Gradients in the Surface after Hard Turning as a Function of Flank Wear and Cutting Speed

**DOI:** 10.3390/ma16041709

**Published:** 2023-02-17

**Authors:** Anna Mičietová, Mária Čilliková, Róbert Čep, Miroslav Neslušan, Nikolaj Ganev

**Affiliations:** 1Faculty of Mechanical Engineering, University of Žilina, Univerzitná 1, 010 26 Žilina, Slovakia; 2Faculty of Mechanical Engineering, VŠB—Technical University of Ostrava, 17. Listopadu 2172/15, 708 00 Ostrava, Czech Republic; 3Faculty of Nuclear Sciences and Physical Engineering, Czech Technical University in Prague, Trojanova 13, 120 00 Prague, Czech Republic

**Keywords:** white layer, heat affected zone, austenite fraction, microhardness, hard turning

## Abstract

This paper investigates the influence of cutting speed and flank wear on the depth profile of residual stresses, as well as the fraction of retained austenite after hard turning of quenched bearing steel 100Cr6. Residual stress and retained austenite profiles were studied for the white layer, heat-affected zone thickness, and XRD sensing depth. It was found that the influence of flank wear on the white layer and heat-affected zone thickness predominates. On the other hand, residual stresses are affected the cutting speed and the superimposing contribution of flank wear. Moreover, these aspects also alter microhardness in the affected regions. The study also demonstrates that information concerning residual stresses and the austenite fraction is integrated into the white layer, and the heat-affected zone in the surface is produced by the insert of low flank wear since the XRD sensing depth is more than the thickness of the white layer. On the other hand, the pure contribution of the white layer or the heat-affected zone to residual stress and the austenite fraction can be investigated when the affected surface region is thick enough.

## 1. Introduction

Hard turning cycles have been widely used for many decades in the real industry for the production of critical components after quenching [1]. Hard turning can be employed for surface finishing [2] or in combination with grinding [3]. The surface state after hard turning is different as compared with hard turning [4]. The main advantages of hard turning as compared with grinding are the ecological benefits of avoiding coolant [5] and/or the machining of part of complex geometry [6]. On the other hand, some disadvantages should also be mentioned, such as the high local stresses in the tool–workpiece interface, the high temperatures in the cutting zone [6], or/and a much steeper stress and structure gradient within the thin near-surface region [7]. Hard turning initiates near-surface refinement [8]. Hard turning surfaces can also suffer from white layer (WL) formation on the near surface [9] and the underlying heat-affected zone (HAZ) being sensitive to the cutting conditions [10]. Such a “sandwich” surface should be avoided in some applications since the brittle and secondary re-hardened WL is lying on the softer HAZ [11]. Flank wear (*VB*) plays a strong role in the thickness of WL and HAZ [12] due to alterations in the stress state and heat transfer [13]. It has already been reported that the presence of WL can lead to early microcracking [14] and premature failures of components [15]. The presence of retained austenite [16] and a very steep stress and microstructure gradient [17] should also be considered as the main sources of near-surface failures.

The surface state during hard turning can also be affected by cooling conditions [18] when MQL and cryogenic cooling is employed. Tool wear and cooling conditions affect the accuracy of produced parts and surface integrity in the complexity of this term [19]. Attanasio et al. [20] studied the influence of cutting speed and feed on WL and HAZ of AISI 52100, including the finite element model integrating the main aspects affecting the surface state. Bosheh and Mativenga [21] investigated the influence of cutting speed on WL in hard-turning H13 tool steel. Alok and Das [22] found that cutting temperature, as well as the thickness of WL, are affected by cutting speed.

The acceptable *VB* values for hard turning cycles are lower than those for conventional turning cycles. However, relatively high *VB* can occur in the production of large components (for example, in the production of bearing rings of diameters up to 4 m). WL and HAZ have already been widely studied by many techniques. Metallographic observations, SEM, TEM, or XRD techniques were employed for a variety of applications in which hard-turned surfaces are produced in order to obtain information about microstructure, austenite fraction, carbides stability and/or the residual stress state [11]. Additionally, the simulation of grain size [3], the predictive models for residual stresses modeling [23], and WL formation [24] were also reported earlier. Mital and Liu et al. [25] developed a special model for the calculation of stresses in super finish hard turning. The XRD technique is mostly integrated into experimental techniques in order to investigate residual stresses and the austenite fraction [26]. Moreover, certain information can be obtained with respect to hardness when crystallite size and micro deformation are calculated [27]. This technique provides reliable information without sophisticated sample preparation. On the other hand, the sensing depth of the XRD technique employing the conventional Cr radiation is very often more than the thickness of the near-surface layer that is affected by hard turning. Therefore, the XRD pattern can integrate information from WL, HAZ, and the untouched bulk. For this reason, information about stress state and/or austenite fraction in the WL or HAZ only cannot usually be separated. This problem can be solved when the low-angle X-ray technique is employed in order to decrease the XRD sensing depth. However, this approach needs very precise etching of the near-surface WL when XRD patterns from HAZ are also required.

The second approach is based on the study of the surface produced by the insert of very high *VB* (0.6 mm or more) when the thickness of WL and HAZ is much more than the XRD sensing depth. Such experiments were already carried out by Guo and Sahni [4]. These authors compared the hard-turned and ground surfaces with carbides stability, WL thickness, microhardness, etc. The contribution of *VB* and cutting speed was also investigated on the quenched parts through the experimental decomposition of cutting forces [28], as well as the time duration of the thermal cycle [29].

This study investigates the influence of cutting speed and *VB* on WL and HAZ (these aspects prevail with surface re-hardening or/and thermal softening). The novelty of this study can be viewed as an assessment of residual stresses and the austenite fraction as a result of the pure contribution of WL or HAZ employing the insert of very high *VB,* which produces a surface of the extreme thickness of WL and underlying HAZ. The study is based on the XRD technique, and the aforementioned aspects of microstructure are supplemented by the measurements of microhardness. As compared with the previously reported papers, the specific character of the surfaces produced by the inserts of very high VB enables the investigation of WL and HAZ properties separately.

## 2. Experimental Setup

The experimental work was carried out using the quenched steel 100Cr6 of hardness 62 ± 1 HRC (quenched from an austenitisation temperature of 840 °C, quenched in an oil temperature of 60 °C, and followed by 2 h tempering at 160 °C). The face turning was run on samples with a diameter of 115 mm and a thickness of 20 mm (2 samples for each cutting condition). The details about the turning process can be found in Table 1. *VB* was measured before and after the cutting process. The views of flank wear land can be found in [28] and in Figure 1. The region of tool wear can be found on the insert’s nose radius. It can be directly linked with the insert geometry as that indicated in Table 1. The images at lower magnifications can be found in the previous study [30]. The measured values of *VB* are listed in Table 2. The inserts of *VB* (as indicated in Table 2) were prepared in the preliminary phase of experiments. Therefore, for each VB, different inserts were employed.

The thickness of WL and HAZ was measured on the metallographic images cut along the cutting direction (see Figure 2). Ten-millimetre-long pieces were cut off, hot moulded, ground, polished, and Nital etched (5% Nital concentration for 10 s). The hot moulded specimens produced by the insert of *VB* = 0.8 mm were, after metallographic observation, also employed for microhardness measurements *HV0.05*. The thicknesses of WL and HAZ for the surfaces produced by the inserts of lower *VB* were too thin for microhardness measurements. *HV0.05* was measured using an Innova Test 400TM (Innovatest, Maastricht, The Netherlands) (load of 50 g for 10 s). The thicknesses of WL, HAZ, and *HV0.05* were obtained by averaging 5 repetitive measurements.

Residual stresses were measured along the cutting direction (CD) and the feed direction (FD). The XRD patterns were measured by Proto iXRD Combo (Proto Manufacturing Ltd., Montreal, Canada) (CrKα radiation in the plane {211}, sensing depth of about 5 μm, scanning angles of ± 39°, Bragg angle of 156°, elastic constants for calculation of stresses of *½s*_2_ = 5.75 TPa^−1^ and *s*_1_ = −1.25 TPa^−1^, and a collimator of the diameter 1 mm). The weight fraction of retained austenite was measured in the planes bcc{211}, fcc{220}, bcc {200}, and fcc{211} (sensing depth of about 4 μm, a collimator of the diameter 2 mm).

## 3. Results of Experiments and Their Discussion

The thickness of WL and HAZ produced by hard turning are mostly affected by *VB* (see Figure 3 and Table 3 and Table 4). WL appears white on the metallographic images as contrasted against the underlying dark HAZ (see Figure 3). WL is produced mainly by the thermal cycle when the turned surface is heated above the austenitising temperature, followed by quite rapid self-cooling [1,7,11]. HAZ is the region in which the temperatures are below the austenitising temperature, and the effect of thermal softening dominates. WL grows with increasing cutting speed at the expense of decreasing HAZ (see Table 3 and Table 4).

The increasing thickness of WL produced by the insert of higher *VB* is due to the increasing energy transformed into the heat, the considered higher temperatures in the tool–workpiece interface, and the longer time period when the turned surface is exposed to the elevated temperatures [13,28,30]. However, Table 3 and Table 4 clearly indicate that the evolution of WL thickness versus *VB* is not straightforward, but WL drops can be found beyond *VB* = 0.4 mm, followed by the steep increase as a result of the altered cutting edge geometry (cutting edge roundness and rake angle) as it was reported earlier [28,29]. The increase in WL thickness versus cutting speed is not fully systematic. Table 3 only demonstrates that WL only tends to grow. On the other hand, the evolution of HAZ (see Table 4) is more systematic (mostly descending tendency). The main reason can be viewed in the fact that HAZ is a product of thermal softening only, whereas WL is a product of synergistic effects of very high local stresses, phase transformation, and superimposing the thermal cycle [22,29].

WL is referred to as a re-hardened matrix of hardness exceeding the hardness produced by heat treatment, whereas the hardness of the thermally softened HAZ is below the bulk [1,4,29]. Table 5 clearly confirms this information. Moreover, it can be measured that the *HV0.05* drops down along with the cutting speed, whereas the evolution of HAZ *HV0.05* exhibits the opposite behaviour. Therefore, the difference in *HV0.05* between WL and bulk and between HAZ and bulk decreases with increasing cutting speed. For this reason, it can be considered that the decreasing time duration of elevated temperatures in the tool–workpiece interface and increased cutting speed plays a major role in this aspect.

Apart from the surface produced by the insert of *VB* = 0.8.mm, the residual stresses in CD (see Figure 4a) are shifted towards the higher magnitude of tensile stresses for the higher *VB* at the lower cutting speed, whereas the systematic behaviour is missing at the higher cutting speeds. Residual stresses are mostly in the tensile region as contrasted against the compressive ones for FD (see Figure 4b). Residual stresses in CD and FD tend to grow versus cutting speed towards the tensile stresses of higher magnitude. This ascension becomes only gentle for the higher *VB,* and the residual stresses in TD drop-down for *VB* = 0.8 mm after the initial steep increase.

The non-systematic evolution of residual stresses of *VB,* as well as the different gradient of the ascend with the cutting speed (or the descending evolution), should be linked with the altered contribution of the “sandwich” microstructure after hard turning. The thickness of WL at lower *VB* is below the sensing depth of the XRD technique (5 μm) (see Table 3). Therefore, the information about the residual stresses originates from WL and HAZ, whereas the pure contribution of WL can be reported for the thicker WL produced by the inserts of higher *VB*. Moreover, the mechanism of residual stresses producing the final state of residual stresses is quite complicated, especially within WL, when the aforementioned combination of mechanical and thermal effects and superimposing the contribution of phase transition is expected [22,29]. Figure 5 demonstrates that the influence of cutting speed on the WL and HAZ thickness is only minor. On the other hand, the influence of cutting speed on residual stress is valuable, especially for the lowest and highest *VB* (see Figure 6).

A more systematic behaviour can be found for the depth extent of residual stresses, see Figure 7. Residual stresses penetrate deeper along with the increasing *VB,* and the maximum compressive stresses are greater. On the other hand, the remarkable relaxation of residual stresses can be found for *VB* = 0.6 mm due to the aforementioned decrease in WL thickness linked with the mentioned alterations in the cutting-edge geometry [28]. This cutting-edge alteration also makes the maximum compressive stresses for *VB* = 0.8 mm lower as compared with that for *VB* = 0.4 mm.

Figure 8b depicts the remarkable contribution of phase transformation into the residual stresses within WL. The increasing *VB* during hard turning tends to shift the near-surface residual stresses toward the tensile region [9]. However, the phase transformation associated with WL formation tends to shift residual stresses into the compressive region. For this reason, the near-surface residual stresses in CD in Figure 8b are below those measured on the boundary between WL and HAZ. The residual stress in CD within WL in Figure 8b is shifted towards the tensile stresses due to the vanishing intensity of the re-hardening effect. Therefore, the intensity of the matrix alterations for phase transformation is the most intensive on the free surface and becomes less pronounced at deeper layers.

Figure 8b also illustrates that near the boundary between WL and HAZ, the evolution of residual stresses changes rapidly, and the increasing evolution is reversed in the deeper region. Figure 8a demonstrates that the initial growth of tensile stresses due to the presence of the thin WL can be found when the thickness of WL is also below the XRD sensing depth. Figure 8 also depicts that the aforementioned evolution of residual stresses can be found mainly in CD. The FD direction seems to be less affected (Figure 8b) or fully unaffected (Figure 8a).

The weight fraction of retained austenite is near the bulk when WL is very thin (far below the XRD sensing depth) and increases along with the increasing thickness of WL (see Figure 9). Apart from the surfaces produced by the insert of *VB* = 0.8 mm, the influence of cutting speed is only minor. However, the strong correlation coefficient between the WL thickness and the austenite fraction, 0.83, can still be found (see Figure 9b). Moreover, Figure 10a demonstrates that the more developed differences in the austenite fraction can be found especially in the free surface when the higher cutting speed produces a surface with a higher fraction of retained austenite (due to accelerated dynamics of the thermal cycle). On the other hand, the depth profiles of retained austenite for all cutting speeds are very similar from the subsurface toward the deeper layers. Figure 10 clearly depicts that the austenite fraction in the WL is much higher than bulk, which drops down from the free surface towards the deeper region. The fraction of retained austenite is below the bulk around the boundary between WL and HAZ when the re-hardening process starts to be replaced by thermal softening. The austenite fraction increases in the deeper regions of HAZ and attains the bulk level at the end of HAZ, see Figure 10b.

Similar behaviour to that for the austenite fraction can be obtained for *FWHM* of XRD patterns. This parameter is usually linked with the hardness and the corresponding dislocation density [27,30]. Figure 11a depicts the remarkably higher *FWHM* on the free surface as compared with the bulk *FWHM* (6.5 ± 0.12). *FWHM* decreases towards the deeper layers and attains the minimum nearby the boundary between WL and HAZ (which is below the bulk). Additionally, *FWHM* grows in the deeper layers of HAZ and attains the bulk *FWHM.* HV measurements provide information about the average microhardness of WL and HAZ, whereas the *FWHM* indicates how the dislocation density and the corresponding hardness are distributed within WL and/or HAZ. Austenite is a much softer phase compared with martensite. For this reason, one might argue that the increasing austenite fraction in WL should contribute to the lower hardness of WL. However, Figure 11b demonstrates that this effect is only minor, and the increasing hardness of re-hardened martensite prevails.

Finally, the results can be summarised regarding *VB* and cutting speed influence on the main analysed parameters. Table 6 clearly indicates that the growing *VB* mostly contributes to the higher thickness of WL and HAZ and the higher austenite fraction. On the other hand, the systematic evolution of residual stresses along with *VB* is mostly missing. The influence of cutting speed is very different since this parameter mostly affects residual stresses. These stresses are shifted from the compressive regions towards the tensile ones or/and their magnitude increase. The influence of cutting speed on WL and HAZ thickness and austenite fraction is limited.

## 4. Conclusions

The main results of this study can be listed as follows:-Measured residual stresses after hard turning are strongly affected by the phase transformation for WL and XRD sensing depth;-Residual stresses in FD are mostly compressive as contrasted against CD;-Cutting speed tends to shift the residual surface stresses towards the tensile region and/or increases the magnitude of tensile stresses;-Higher *VB* increases the penetration depth of residual stresses.-Phase transformation tends to shift the residual stresses towards the compressive region;-The evolution of residual stresses is reversed on the boundary between WL and HAZ;-The austenite fraction increases along with the increasing thickness of WL;-The austenite fraction in the near-surface WL is increased, whereas it is decreased as compared with the bulk in the HAZ and near the boundary between WL and HAZ;-*FWHM* of XRD and the corresponding microhardness of WL and HAZ are driven by martensite dislocation density, and the contribution of retained austenite is only minor.

The experimental approach in which WL and HAZ are investigated employing the inserts of very high *VB* is promising. The findings of residual stresses evolution (their depth profiles) and retained austenite can be transferred into the hard-turned surface when the thickness of WL is below the sensing depth of the XRD technique. The findings of this study can be employed for the detection of reasons or risk factors associated with unwanted component failures in operation. The main risk factor is linked with the presence of hard and brittle surface WL lying on the softened underlying HAZ. Such a “sandwich” microstructure gradient makes components finished by hard turning quite sensitive to microcracking in the brittle WL under load in the real operation. Moreover, this microcracking can propagate through the WL and towards the deeper regions due to the weakened abutment of thermally softened HAZ.

## Figures and Tables

**Figure 1 materials-16-01709-f001:**
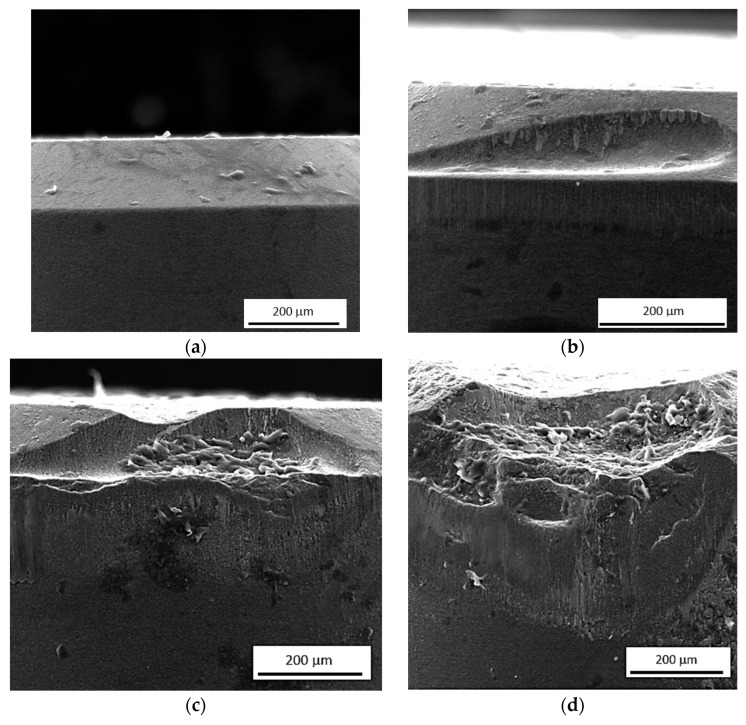
SEM images of worn cutting edges: (**a**) *VB* = 0 mm; (**b**) *VB* = 0.1 mm; (**c**) *VB* = 0.2 mm; (**d**) *VB* = 0.4 mm; (**e**) *VB* = 0.6 mm; (**f**) *VB* = 0.8 mm.

**Figure 2 materials-16-01709-f002:**
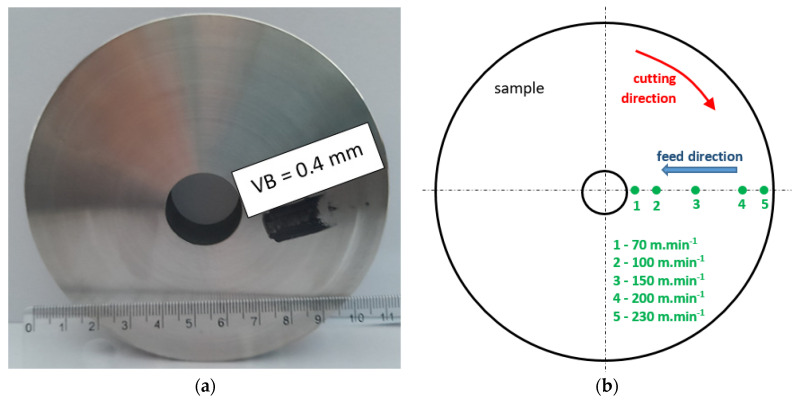
Photo and brief view of the analysed positions, the corresponding cutting speeds on the disk, and the process kinematics: (**a**) photo; (**b**) brief sketch.

**Figure 3 materials-16-01709-f003:**
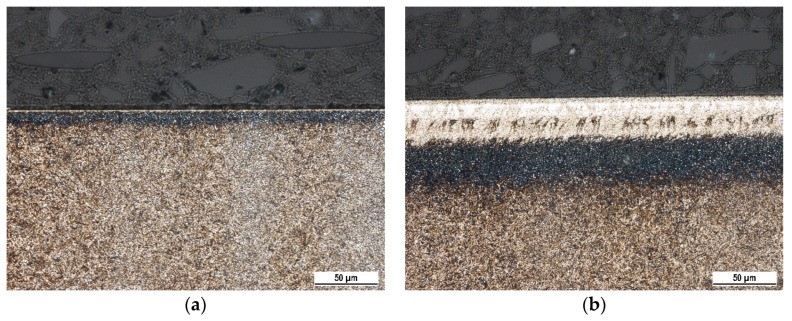
Metallographic images of hard-turned surfaces, *v_c_* = 100 m.min^−1^: (**a**) *VB* = 0.1 mm; (**b**) *VB* = 0.8 mm.

**Figure 4 materials-16-01709-f004:**
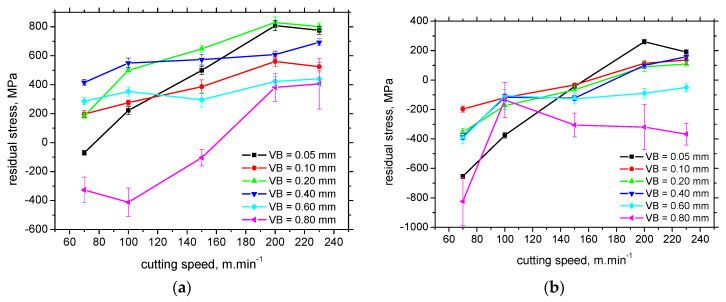
Evolution of surface residual stresses with cutting speed. (**a**) CD, (**b**) FD.

**Figure 5 materials-16-01709-f005:**
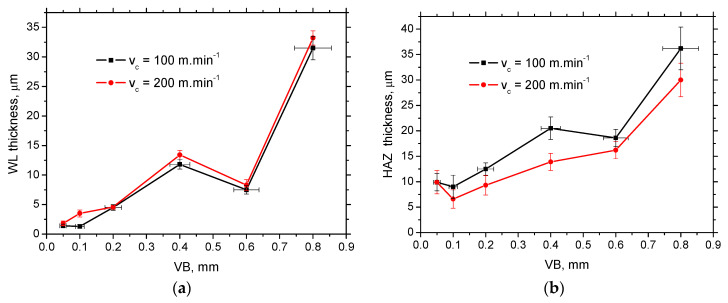
*VB* versus WL and HAZ thickness: (**a**) *VB* versus WL thickness; (**b**) *VB* versus HAZ thickness.

**Figure 6 materials-16-01709-f006:**
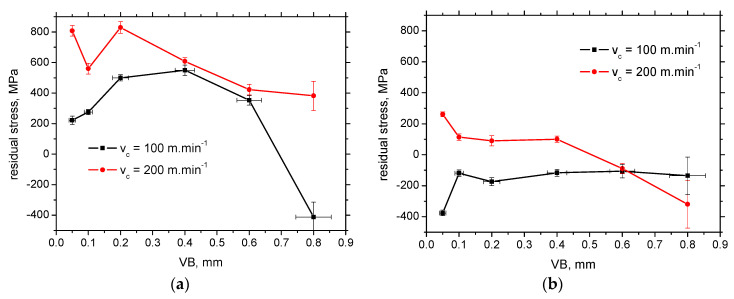
Evolution of surface residual stresses with *VB*: (**a**) CD; (**b**) FD.

**Figure 7 materials-16-01709-f007:**
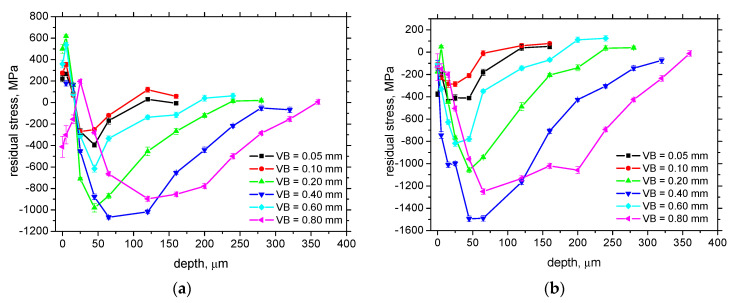
Depth profiles of residual stresses for cutting speed 100 m.min^−1^ (bulk stress 67 ± 12 MPa): (**a**) CD; (**b**) FD.

**Figure 8 materials-16-01709-f008:**
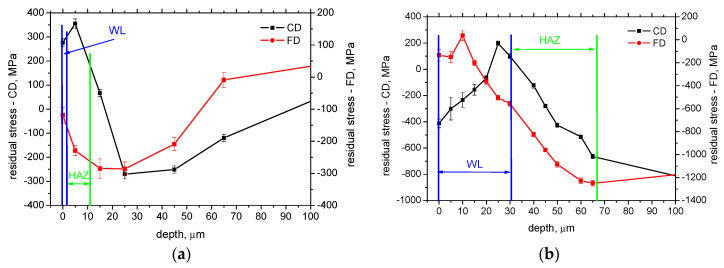
Details of residual stress depth profiles for cutting speed 100 m.min^−1^. (**a**) *VB* = 0.1 mm, (**b**) *VB* = 0.8 mm.

**Figure 9 materials-16-01709-f009:**
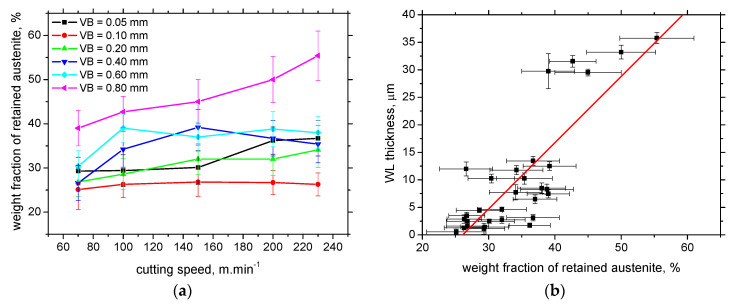
Evolution of surface austenite fraction and its correlation with WL thickness (bulk austenite fraction 22 ± 3.2): (**a**) surface austenite fraction along cutting speed; (**b**) austenite fraction versus WL thickness (correlation coefficient 0.83).

**Figure 10 materials-16-01709-f010:**
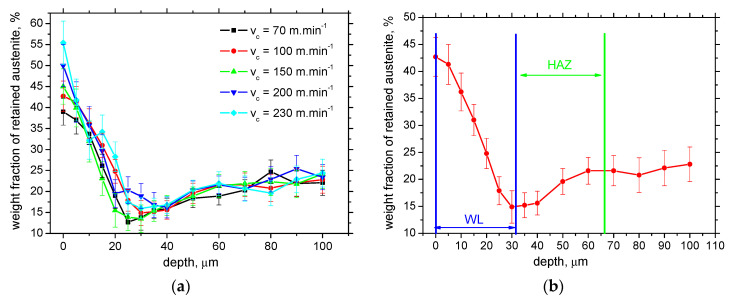
Depth profiles of austenite fraction for *VB* = 0.8 mm and detail of austenite fraction for cutting speed 100 m.min^−1^: (**a**) depth profiles for *VB* = 0.8 mm; (**b**) detail of austenite fraction for cutting speed 100 m.min^−1^.

**Figure 11 materials-16-01709-f011:**
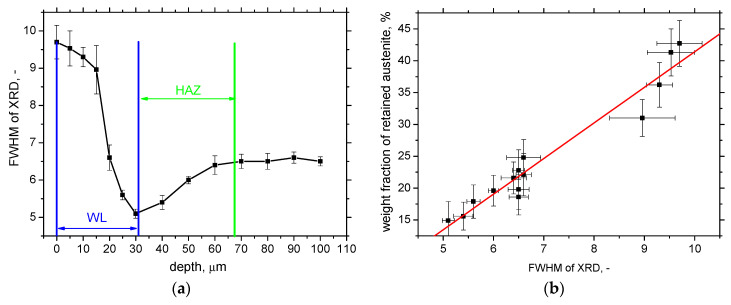
Depth profile of *FWHM* for *VB* = 0.8 mm and cutting speed 100 m.min^−1^ and its correlation with austenite fraction (bulk *FWHM* 6.5 ± 0.12): (**a**) depth profile of *FWHM*; (**b**) *FWHM* versus austenite fraction (correlation coefficient 0.97).

**Table 1 materials-16-01709-t001:** Cutting inserts and cutting conditions.

Cutting insert	DNGA 150408, PCBN, 70% of CBN, TiN coated
Inset geometry	*r_ε_* = 0.8 mm, chamfer (rake) angle −35° of width 250 μm, entering angle 93°
Feed *f*	kept constant at 0.09 mm.rev^−1^
Cutting depth *a_p_*	kept constant at 0.25 mm
Cutting speed *v_c_*	in the range from 70 to 230 m.min^−1^; see Figure 1

**Table 2 materials-16-01709-t002:** Flank wear *VB* of employed inserts.

Average *VB*, mm	0.05	0.1	0.2	0.4	0.6	0.8
*VB* at the beginning of the test, mm	0.04	0.087	0.175	0.37	0.562	0.745
*VB* at the end of the test, mm	0.06	0.113	0.225	0.43	0.638	0.855

**Table 3 materials-16-01709-t003:** Thickness of WL as a function of cutting speed and flank wear *VB* (thickness of WL in μm).

*v_c_*	*VB* = 0.05 mm	*VB* = 0.10 mm	*VB* = 0.20 mm	*VB* = 0.40 mm	*VB* = 0.60 mm	*VB* = 0.80 mm
70 m.min^−1^	**1.3 ± 1.0**	**0.6 ± 0.5**	**2.5 ± 0.4**	12 ± 1.2	10.3 ± 0.8	29.8 ± 3.2
100 m.min^−1^	**1.4 ± 0.3**	**1.3 ± 0.3**	**4.5 ± 0.5**	11.8 ± 0.8	7.5 ± 0.7	31.5 ± 2.0
150 m.min^−1^	**2.5 ± 0.4**	**1.6 ± 0.4**	**2.8 ± 0.6**	12.5 ± 0.8	6.5 ± 0.8	29.5 ± 0.6
200 m.min^−1^	**1.8 ± 0.4**	**3.5 ± 0.6**	**4.6 ± 0.4**	13.4 ± 0.8	8.3 ± 0.9	33.2 ± 1.2
230 m.min^−1^	**3.1 ± 0.5**	**2.9 ± 0.7**	7.7 ± 1.3	10.2 ± 1.0	8.5 ± 0.9	35.8 ± 1.0

Note: when the thickness of WL does not exceed the XRD sensing depth, the numbers are highlighted in red colour.

**Table 4 materials-16-01709-t004:** Thickness of HAZ as a function of cutting speed and flank wear *VB* (thickness of HAZ in μm).

*v_c_*	*VB* = 0.05 mm	*VB* = 0.10 mm	*VB* = 0.20 mm	*VB* = 0.40 mm	*VB* = 0.60 mm	*VB* = 0.80 mm
70 m.min^−1^	8.8 ± 1.7	9.8 ± 1.7	14.0 ± 1.3	21.7 ± 1.7	20.2 ± 2.2	37.8 ± 2.0
100 m.min^−1^	9.9 ± 1.7	9.0 ± 2.3	12.5 ± 1.2	20.5 ± 2.2	18.6 ± 1.7	36.2 ± 4.2
150 m.min^−1^	10.5 ± 0.6	7.0 ± 1.5	10.8 ± 1.5	17.8 ± 2.7	17.5 ± 0.9	32.5 ± 3.0
200 m.min^−1^	9.9 ± 2.3	6.6 ± 1.8	9.3 ± 1.9	13.9 ± 1.7	16.2 ± 1.7	29.9 ± 3.3
230 m.min^−1^	8.8 ± 0.8	7.5 ± 0.8	8.0 ± 1.0	8.8 ± 1.1	13.5 ± 0.9	27.5 ± 2.7

**Table 5 materials-16-01709-t005:** *HV0.05* for WL and HAZ as a function of cutting speed, *VB* = 0.8 mm, bulk *HV0.05* 726 ± 10.

*v_c_*	WL	HAZ
70 m.min^−1^	995 ± 23	638 ± 34
100 m.min^−1^	953 ± 25	641 ± 29
150 m.min^−1^	930 ± 29	662 ± 22
200 m.min^−1^	920 ± 35	670 ± 26
230 m.min^−1^	848 ± 37	683 ± 16

**Table 6 materials-16-01709-t006:** Influence of *VB* and cutting speed on residual stresses, austenite fraction, and WL and HAZ thickness.

	WL	HAZ	Stress CD	Stress FD	Austenite
*VB*	↑	↑	-	-	↑
cutting speed	-	-	↑*	↑*	-

Legend: ↑—increasing tendency, -—non-systematic behaviour, ↑*—shift towards the tensile stresses or/and their higher magnitude.

## Data Availability

The raw data required to reproduce these findings cannot be shared easily due to technical limitations (some files are too large). However, authors can share the data on any individual request (please contact the corresponding author by the use of its mailing address).

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
