# Peer review of "Study of Residual Stresses and Austenite Gradients in the Surface after Hard Turning as a Function of Flank Wear and Cutting Speed"

_materials, 2023, doi:10.3390/ma16041709_

Round 1
Reviewer 1 Report
Only minor typos.
Line 94 space between 10 s
Line 95 & 160 0,8.mm -- > 0,8 mm
Author Response
Reviewer n. 1:
All changes made in the manuscript (additional texts and corrections) are highlighted yellow colour (valid for the manuscript as well as this document).
Reviewer: Line 94 space between 10 s
Response: corrected
Manuscript:
HV0.05 was measured using an Innova Test 400TM (load of 50 g for 10 s).
Reviewer: Line 95 & 160 0,8.mm -- > 0,8 mm
Response: corrected
Manuscript:
The hot moulded specimens produced by the insert of VB = 0.8 mm were, after …
Reviewer 2 Report
Line 39: Why do you name VB to flank wear?, what does this term mean?
Author Response
Reviewer: Why do you name VB to flank wear?, what does this term mean?
Response: It is generally accepted abbreviation for flank wear. This abbreviation originates from German language.
Manuscript: we prefer no change.
Reviewer 3 Report
Manuscript Number: materials-2187057
Title: Study of residual stresses and austenite gradients in the surface after hard turning as a function of flank wear and cutting speed
Decision: Major revision
Article Type: Article
The article is, in general, well written but there are some issues that article should consider to revise in order to improve its quality. Some comments were done in this way:
Ø In the summary (line 17), the net rate should be given instead of “the contribution of the cutting speed is only minor”.
Ø Table 2 is not fully understood. Please choose a different method.
Ø Images of wearing cutting tools should also be in this study. If the previous work given in source 25 was given as Part I, it would be acceptable, but there is quite a long time and differences between the two studies. Please consider this work as a new work and make it easier for the readers. You also said on line 112 that WL was affected by Vb. If the subject of the article is the decrease of WL, Vb must be given as fig.
Ø Please give Fig 1 in 3D.
After making the above corrections would recommend this article for publication in Materials.
Author Response
Reviewer n. 3:
All changes made in the manuscript (additional texts and corrections) are highlighted yellow colour (valid for the manuscript as well as this document).
Reviewer: In the summary (line 17), the net rate should be given instead of “the contribution of the cutting speed is only minor”.
Response: The reviewer proposed substitution is quite debatable for us. The term “cutting speed” is fully clear. The proposed substitution could result in the sentence meaning misunderstanding.
Manuscript: We prefer no change.
Reviewer: Table 2 is not fully understood. Please choose a different method.
Response: We altered the appearance of the table 2. Now it is clearer.
Manuscript: Please check appearance of table 2.
Reviewer: Images of wearing cutting tools should also be in this study. If the previous work given in source 25 was given as Part I, it would be acceptable, but there is quite a long time and differences between the two studies. Please consider this work as a new work and make it easier for the readers.
Reviewer: You also said on line 112 that WL was affected by Vb. If the subject of the article is the decrease of WL, Vb must be given as fig.
Response: We added images of flank wear land.
Manuscript: Please check new Figure 1.
Reviewer: Please give Fig 1 in 3D.
Response: we added photo of the sample.
Manuscript:
please check appearance of Fig. 2.
Reviewer 4 Report
1. A photo of the workpiece should be provided next to Fig.1 with dimensions labelled.
2. How is VB measured and controlled? It should keep increasing during the turning. What are the changes before and after the cutting? Does Table 2 provide average flank wear? 3. Could you show photos of tools with different flank wears? It would be good to have one from [25] with proper citation. 4. What is the main motivation to select VB as an indicator? It is a measured parameter that cannot be directly controlled in reality. 5. In conclusion, it mentions that the work can help detect the risk factors. Could you be more specific and give more quantitative conclusions?
Author Response
Reviewer n. 4:
All changes made in the manuscript (additional texts and corrections) are highlighted yellow colour (valid for the manuscript as well as this document).
Reviewer: A photo of the workpiece should be provided next to Fig.1 with dimensions labelled.
Response: we added photo of the sample.
Manuscript:
please check appearance of Fig. 2.
Reviewer: How is VB measured and controlled? It should keep increasing during the turning. What are the changes before and after the cutting? Does Table 2 provide average flank wear?
Response: we prepared the insert of VB indicated in table 2 in the preliminary phase. For each indicated VB the different insert was employed. VB was measured before experiment as well as after hard turning. Table 2 indicates these values together with average VB, see altered table 2.
Manuscript:
VB was measured before and after the cutting process. The views of flank wear land can be found in [25] as well as in Figure 1. The measured values of VB are listed in Table 2. The inserts of VB (as that indicated in Table 2) were prepared in the preliminary phase of experiments. Expressed in other words, for each VB the different inserts was employed.
Reviewer: Could you show photos of tools with different flank wears? It would be good to have one from [25] with proper citation
Response: We added images of flank wear land.
Manuscript: Please check new Figure 1.
Reviewer: What is the main motivation to select VB as an indicator? It is a measured parameter that cannot be directly controlled in reality.
Response: Influence of cutting depth, feed and cutting speed on WL and HAZ is less as contrasted against VB. That is the reason why we investigate WL and HAZ mostly as a function of VB. However, we also investigated contribution of cutting speed in our study. Our results indicate that the influence of VB prevails.
VB in the real industrial conditions is can be easily linked with cutting time. That is the indirect way (together for example measurement of workpiece size) to have certain information about developed VB.
Manuscript: we added note.
This study investigates the influence of cutting speed as well as VB on WL and HAZ (these aspects prevails with respect of surface re-hardening or/and thermal softening).
Reviewer: In conclusion, it mentions that the work can help detect the risk factors. Could you be more specific and give more quantitative conclusions?
Response: We are not capable to add quantitated conclusion with respect of risky factor associated with hard turning. The main risky factor is linked with presence of hard and brittle surface WL lying on the softened underlying HAZ. Such “sandwich” microstructure gradient makes components finished by hard turning quite sensitive to the microcracking in the brittle WL under load in the real operation. Moreover, this microcracking can propagate through the WL as well as towards the deeper regions due to weakened abutment of thermally softened HAZ.
Manuscript: We added this statement.
The main risky factor is linked with the presence of hard and brittle surface WL lying on the softened underlying HAZ. Such “sandwich” microstructure gradient makes components finished by hard turning quite sensitive to microcracking in the brittle WL under load in the real operation. Moreover, this microcracking can propagate through the WL as well as towards the deeper regions due to weakened abutment of thermally softened HAZ.
Reviewer 5 Report
The introduction needs to be completed. The authors have not presented the state of the art on the influence of cutting tool wear and cutting speed on the state of the surface layer. Please carry out an analysis of the articles on the state of the surface layer after hard turning.
Furthermore, please explain why the effect of cutting speed is being investigated and no other parameters such as feed rate or depth of cut.
Please indicate clearly what is new in the article, what is new knowledge.
Table 1: Please describe in detail the grade of cutting material used, include full composition. Please specify full insert geometry. Please specify cutting tool geometry, for example: rake angle, entering angle. Please define feed in appropriate unit.
Please explain table 2, what means n?
Please describe in detail the study, how many turning tests were performed.
The experiment was incorrectly designed. In one face turning test, the authors assumed a constant value for VB tool wear. In reality, VB wear varied along the cutting path, by up to 50%. Therefore, it is not possible to clearly state whether the changes in the surface layer are only due to the cutting speed or only to VB wear. Please correct or clarify this.
The authors concluded that the thickness of WL and HAZ produced by hard turning are mostly affected by VB. But there is no graph as a function of VB in the article. Please add graphs of WL, HAZ and stress variation as a function of VB in addition to cutting speed.
Please develop experimental models from the results obtained and compare them. Please perform a statistical analysis of the results.
Author Response
Reviewer n. 5:
All changes made in the manuscript (additional texts and corrections) are highlighted yellow colour (valid for the manuscript as well as this document).
Reviewer: The introduction needs to be completed. The authors have not presented the state of the art on the influence of cutting tool wear and cutting speed on the state of the surface layer. Please carry out an analysis of the articles on the state of the surface layer after hard turning.
Response: We consider that our list of references with respect of WL and HAZ is rich enough. But we added additional papers associated with the studied field.
Manuscript:
Surface state during hard turning can be also affected by cooling conditions [19] when MQL and cryogenic cooling is employed. Tool wear and cooling conditions affect accuracy of produced parts and surface integrity in the complexity of this term [20]. Attanasio et. al [21] studied influence of cutting speed and feed on WL and HAZ of AISI 52100 including finite elements model integrating the main aspects affecting surface state. Bosheh and Mativenga [22] investigated influence of cutting speed on WL in hard turning H13 tool steel. Alok and Das [23] found that cutting temperature as well as the thickness of WL are affected by cutting speed.
Reviewer: Furthermore, please explain why the effect of cutting speed is being investigated and no other parameters such as feed rate or depth of cut.
Response: Influence of cutting depth and feed on WL and HAZ is less as contrasted against especially VB. Also cutting speed affects mostly stress state and its influence of WL and HAZ thickness is less. That is the reason why we investigate WL and HAZ mostly as a function of VB and cutting speed. We also carried out investigation of feed on life time and related aspects in the past but influence of feed is only limited.
Manuscript: we added note.
This study investigates the influence of cutting speed as well as VB on WL and HAZ (these aspects prevails with respect of surface re-hardening or/and thermal softening).
Reviewer: Please indicate clearly what is new in the article, what is new knowledge.
Response: This information has been already indicated in the “Introduction” part. However, we added further explanation at the end of “Introduction”.
Manuscript:
The novelty of this study can be viewed as an assessment of residual stresses and the austenite fraction as a result of the pure contribution of WL or HAZ employing the insert of very high VB which produces a surface of the extreme thickness of WL and underlying HAZ. The study is based on the XRD technique, and the aforementioned aspects of microstructure are supplemented by the measurements of microhardness. As compared with the previously reported papers, the specific character of the surfaces produced by the inserts of very high VB enables investigation of WL as well as HAZ properties separately.
Reviewer: Table 1: Please describe in detail the grade of cutting material used, include full composition. Please specify full insert geometry. Please specify cutting tool geometry, for example: rake angle, entering angle. Please define feed in appropriate unit.
Response: we added information about rake and entering angles, other parameters have been already listed in the first version of the Table 1. We have no further information about insert geometry.
Feed is in mm. It is correct.
Manuscript: check Table 1.
Reviewer: Please explain table 2, what means n?
Response: We altered the appearance of the table 2. Now it is clearer.
Manuscript: Please check appearance of table 2.
Reviewer: Please describe in detail the study, how many turning tests were performed.
Response: we carried out turning of 2 pieces for each conditions. The surface state was very similar.
Manuscript:
The face turning was run on samples with a diameter 150 mm and a thickness 20 mm (2 samples for each cutting conditions).
Reviewer: The experiment was incorrectly designed. In one face turning test, the authors assumed a constant value for VB tool wear. In reality, VB wear varied along the cutting path, by up to 50%. Therefore, it is not possible to clearly state whether the changes in the surface layer are only due to the cutting speed or only to VB wear. Please correct or clarify this.
Response: Reviewer opinion is right. For this reason, we altered concept of the experiments and we added this information. We prepared the insert of VB indicated in table 2 in the preliminary phase. For each indicated VB the different insert was employed. VB was measured before experiment as well as after hard turning. Table 2 indicates these values together with the average VB, see altered table 2. We employed PCBN inserts. Evolution of wear of these inserts with time is remarkably slower as compared with ceramic inserts. Being so, such experiment provides acceptable stability of cutting conditions as well as VB in a certain range. Please check also the appearance of Table 2.
Manuscript:
VB was measured before and after the cutting process. The views of flank wear land can be found in [25] as well as in Figure 1. The measured values of VB are listed in Table 2. The inserts of VB (as that indicated in Table 2) were prepared in the preliminary phase of experiments. Expressed in other words, for each VB the different inserts was employed.
Reviewer: The authors concluded that the thickness of WL and HAZ produced by hard turning are mostly affected by VB. But there is no graph as a function of VB in the article. Please add graphs of WL, HAZ and stress variation as a function of VB in addition to cutting speed.
Response: we added required figures.
Manuscript: please see new figures 5, 6. We also added the corresponding text.
Figure 5 demonstrates that influence of cutting speed on the WL as well as HAZ thickness in only minor. On the other hand, the influence of cutting speed on residual stress is valuable especially for the lowest and highest VB (see Figure 6).
Reviewer: Please develop experimental models from the results obtained and compare them.
Response: This note is very good. We have developed the summarisation of our results in systematic manner. We analysed influence of input parameters such as cutting speed and VB on HAZ and WL thickness, austenite fraction as well as residual stresses. Instead a model or models we edited the new Table 6 as well as the corresponding text. Please check the manuscript – the end of experimental part. Might be this would be satisfactory enough.
Manuscript:
Table 6. Influence of VB and cutting speed on residual stresses, austenite fraction as well as WL and HAZ thickness.
|
|
WL |
HAZ |
Stress CD |
Stress FD |
Austenite |
|
VB |
↑ |
↑ |
- |
- |
↑ |
|
cutting speed |
- |
- |
↑* |
↑* |
- |
Legend:
↑ - increasing tendency, - non-systematic behaviour, ↑* - shift towards the tensile stresses or/and their higher magnitude.
Finally, the results can be summarized with respect of VB and cutting speed influence on the main analysed parameters. Table 6 clearly indicates that the growing VB mostly contributes to the higher thickness of WL and HAZ as well as the higher austenite fraction. On the other hand, systematic evolution of residual stresses along with VB is mostly missing. Influence of cutting speed is very different since this parameter mostly affects residual stresses. These stresses are shifted from compressive region towards the tensile ones or/and their magnitude increase. Influence of cutting speed on WL and HAZ thickness as well as austenite fraction is limited.
Reviewer: Please perform a statistical analysis of the results.
Response: We performed statistical analysis of our results in the original version of manuscript. We are not capable to remarkably improve this aspect.
We measured VB at the begin as well as at the end of the test – Table 2.
Tables 3 and 4 indicate STDV of repetitive measurements of WL and HAZ from 5 measurements.
Table 5 indicates STDV of repetitive measurements of HV from 5 measurements.
Fig. 4 indicates STDV of measured stresses.
Fig. 5 provides STDV for thickness and range of VB.
Fig. 6 provides STDV for stresses and range of VB.
Fig. 7 and 8 indicate STDV of measured stresses.
Fig. 9a indicates STDV of measured austenite and we also calculated correlation coefficient on the base of data in Fig.9b.
Fig. 10 indicates STDV of measured austenite.
Fig. 11a indicates STDV of measured austenite and we also calculated correlation coefficient on the base of data in Fig. 11b.
Manuscript: We prefer no change since our statistical information originated from repetitive measurements is huge enough. We consider that we did our best and all required statistical calculations together with correlation analyses are good enough.
Round 2
Reviewer 3 Report
Some fixes are missing. The following issues should be clarified.
-) The authors used the word cutting speed.
"It was found that the influence of flank wear with respect to the white layer and heat-affected zone thickness predominates, whereas the contribution of the cutting speed is only minor. On the other hand, residual stresses are affected the cutting speed and the superimposing contribution of flank wear." This sentence is your sentence.
The expression "only minor" here should be given numerically or completely removed.
-) "Response: We added images of flank wear land." Flank wear images must be added separately for all cutting tools. In addition, the SEM images given in Fig 1 should be taken from further away and the location of the cutting tool should be detailed.
Author Response
Reviewer: "It was found that the influence of flank wear with respect to the white layer and heat-affected zone thickness predominates, whereas the contribution of the cutting speed is only minor. On the other hand, residual stresses are affected the cutting speed and the superimposing contribution of flank wear." This sentence is your sentence.
The expression "only minor" here should be given numerically or completely removed.
Response: We removed the asked part of the sentence.
Manuscript:
"It was found that the influence of flank wear with respect to the white layer and heat-affected zone thickness predominates. On the other hand, residual stresses are affected the cutting speed and the superimposing contribution of flank wear."
Reviewer: "Response: We added images of flank wear land." Flank wear images must be added separately for all cutting tools.
Response: We added next 2 SEM images for new tool and insert of VB = 0.1 mm.
Manuscript: Please check appearance of Fig. 1a,b.
Reviewer: In addition, the SEM images given in Fig 1 should be taken from further away and the location of the cutting tool should be detailed.
Response: The images at lower magnifications can be found in our previous study. Please check our manuscript [30]. Tool wear is located in the region of cutting insert nose radius. This information can be easily linked with insert geometry given in Table 1.
Manuscript: Region of tool wear can be found on the inserts nose radius. It can be directly linked with the insert geometry as that indicated in Table 1. The images at lower magnifications can be found in the previous study [30].
Reviewer 4 Report
Comments are addressed properly.
Author Response
Reviewer: Comments are addressed properly.
Response: no response
Manuscript: no change
Reviewer 5 Report
Thank you very much for your responses and improving the article. However, I have one comment that should be addressed. I disagree with the answer that the feed parameter is defined in millimeters. In every scientific article on machining, feed is defined in mm/rev, mm/tooth or mm/min. It is a parameter describing motion so it cannot be defined in a unit of length (mm) because it makes no sense. You must always specify the value of the displacement in the feed motion per revolution, time or tooth.
Please see how feed rate is defined in published and peer-reviewed articles:
https://doi.org/10.3390/ma16031205
https://doi.org/10.3390/ma16030949
Author Response
Reviewer: Thank you very much for your responses and improving the article. However, I have one comment that should be addressed. I disagree with the answer that the feed parameter is defined in millimeters. In every scientific article on machining, feed is defined in mm/rev, mm/tooth or mm/min. It is a parameter describing motion so it cannot be defined in a unit of length (mm) because it makes no sense. You must always specify the value of the displacement in the feed motion per revolution, time or tooth.
Response: We altered units for feed as indicated.
Manuscript: please check unit of feed in table 1
kept constant 0.09 mm.rev-1